# Reproducibility study - Does enforcing diversity in hidden states of LSTM-Attention models improve transparency?

## Reproducibility Summary

It has been shown (Jain and Wallace, 2019) that the weights in attention mechanisms do not necessarily offer a faithful explanation of the model's predictions. In the paper 'Towards Transparent and Explainable Attention Models' (Mohankumar et al., 2020) the authors propose two methods to enhance faithfulness and plausibility of the explanations provided by an LSTM model combined with a basic attention mechanism.

**Scope of Reproducibility**    For this reproducibility study, we focus on the main claims made in this paper:

- The attention weights in standard LSTM attention models do not provide faithful and plausible explanations for its predictions. This is potentially because the conicity of the LSTM hidden vectors is high.
- Two methods can be applied to reduce conicity: Orthogonalization and Diversity Driven Training. When applying these methods, the resulting attention weights offer more faithful and plausible explanations of the model's predictions, without sacrificing model performance.

**Methodology**    The paper includes a link to a repository with the code used to generate its results. All our experiments with this code are conducted on GPU nodes of the Lisa Cluster at SURFsara[1]. We follow four investigative routes: (i) Replication: we rerun experiments on datasets from the paper in order to replicate the results, and add the results that are missing in the paper; (ii) Code review: we scrutinize the code to validate its correctness; (iii) Evaluation methodology: we extend the set of evaluation metrics used in the paper with the LIME method, in an attempt to resolve inconclusive results; (iv) Generalization to other architectures: we test whether the authors' claims apply to variations of the base model (more complex forms of attention and a BiLSTM encoder)

**Results**    We confirm that the Orthogonal and Diversity LSTM achieve similar accuracies as the Vanilla LSTM, while lowering conicity. However, we cannot reproduce the results of several of the experiments in the paper that underlie their claim of better transparency. In addition, a close inspection of the code base reveals some potentially problematic inconsistencies. Despite this, under certain conditions, we do confirm that the Orthogonal and Diversity LSTM can be useful methods to increase transparency. How to formulate these conditions more generally remains unclear and deserves further research. The single input sequence tasks appear to benefit most from the methods. For these tasks, the attention mechanism does not play a critical role for achieving performance.

**What was easy / difficult**    The codebase of the authors is accessible and can be run easily, with good facilities to prepare datasets and define configurations. The Orthogonalization and Diversity Training methods are well explained in the paper and mostly cleanly implemented. The larger datasets (Amazon and CNN) are difficult to run due to memory requirements and compute times. The codebase can be hard to navigate, a consequence of the choice to accommodate a large variation of models and datasets in one framework.

**Communication with original authors**    We reached out to the authors on a fundamental but unexplained choice in the model architecture but unfortunately did not hear back before the deadline of our assignment.

---

[1]We had access to two Nvidia GTX1080Ti/11Gb VRAM GPUs. `https://userinfo.surfsara.nl/systems/lisa/description`.

# 1 Introduction

The popularity of attention models has sparked many studies on the interpretability of the attention distributions, with often conflicting claims (Jain and Wallace, 2019; Wiegreffe and Pinter, 2019; Serrano and Smith, 2019). Mohankumar et al. (2020) argue that the reason why attention weights do not always provide a faithful explanation of the model's predictions is that the learned hidden states of the LSTM based encoder are very similar across time steps, which is expressed by high conicity of these vectors. As a result, random permutation of the attention weights leads to a similar final context vector, which implies the weights do not provide a faithful explanation. The authors propose two methods that force the hidden states of the LSTM to be more diverse. Orthogonal LSTM ensures low conicity by orthogonalizing the hidden state at time $t$ with respect to the mean of the previous hidden states. In Diversity LSTM the model is trained to jointly maximize the log-likelihood of the training data and to minimize the conicity of the hidden states.

# 2 Scope of reproducibility

In this reproducibility study we focus on the authors' main claim that the Diversity LSTM and Orthogonal LSTM lead to more faithful and plausible explanations, while maintaining accuracy of the predictions. The authors support their claim by evaluating a series of metrics (Mohankumar et al., 2020) that are assumed to be indicative of levels of faithfulness and plausibility. We follow four investigative routes:

- Replication: The main part of our study is focused on reproducing the results on the metrics in Mohankumar et al. (2020), and to validate whether we can confirm their observations and conclusions. Furthermore, as the original paper only presents the results of a selection of models and datasets, we complement the results where possible. Most notably, we add results on the Orthogonal LSTM that were not in the original paper. Models, code and datasets are described in Section 3. Our replication results are presented in Section 4;

- Code review: As the authors' code[2] is publicly available, we use their code for the reproduction. In Section 5 we investigate whether the implementation is consistent with the description of the algorithms in the paper;

- Evaluation methodology: In Section 6 we report on our attempt to resolve inconclusive results we found on the attribution methods by extending the set of evaluation metrics used in the paper with the LIME method[3];

- Generalization to other architectures: In Section 7 we test whether the authors' claims apply to variations of the base model (more complex forms of attention and a BiLSTM encoder).

We conclude this paper in Section 8 with a discussion on the conditions under which the proposed methods are most likely to be effective, and a reflection on our replication study.

# 3 Methodology

**Code** The code accompanying the paper is an extension based on the code first developed by Jain and Wallace (2019)[4]. The entry point of the code is clear and well documented and allows a user to define specific jobs using command line arguments for hyperparameters. Preprocessing routines for the most datasets are included[5].

**Datasets** We reran the experiments on 11 of the 14 datasets used in the paper. The nature and size of the datasets covers a wide range, from relatively simple binary sentiment classification tasks with single input sequence (abbreviated: SS) (e.g. SST with average input sentence length of 20 words), to complex question answering tasks with dual input sequences (abbreviated: DS)[6] (e.g. CNN with average document size of 760 words and an average of 26 answer categories). Some illustrations of data points can be found in Appendix D. The code repository includes links to the datasets, as well as the pre-processing routines used by the authors. We excluded the Amenia and Diabetes datasets because they were not accessible. The Amazon dataset caused memory issues when running the experiments. Despite these issues we were able to get the accuracies and conicity values for this dataset.

**Model descriptions** The baseline model (Vanilla LSTM) used in the paper is shown in Figure 1. For DS tasks, it consists of two uni-directional LSTM encoders that act on a P-path (for document input phrases) and a Q-path (for

---

[2] `https://github.com/akashkm99/Interpretable-Attention`

[3] The code with all extensions we made for this review can be accessed at `https://anonymous.4open.science/r/FACT_AI_project/`

[4] `https://github.com/successar/AttentionExplanation`

[5] Diabetes and Anemia are not included. These datasets are not publicly available (ethics screening required before use)

[6] These distinctions are differently named in the code: SS is referred to as `BC` and DS as `QA`

question input phrases). When applied on SS tasks in the paper, only the P-path is used. An attention decoder is applied to the hidden states of the P-path LSTM to form the context vector $\mathbf{c}_\alpha$ on which the model calculates its output. The last hidden state of the Q-path is used as the query term for DS tasks.

The Diversity and Orthogonal LSTM that Mohankumar et al. (2020) propose are variants of the baseline model. The **Orthogonal LSTM** applies an orthogonalization procedure to the LSTM hidden state vectors during training: the hidden state in timestep $t$ is set to the component that is orthogonal to the mean of previous hidden states. This enforces low conicity of the hidden state vectors $\mathbf{h}_t^p$. The **Diversity LSTM** uses a standard LSTM cell with no explicit orthogonalization, but minimizes conicity jointly with the standard loss.

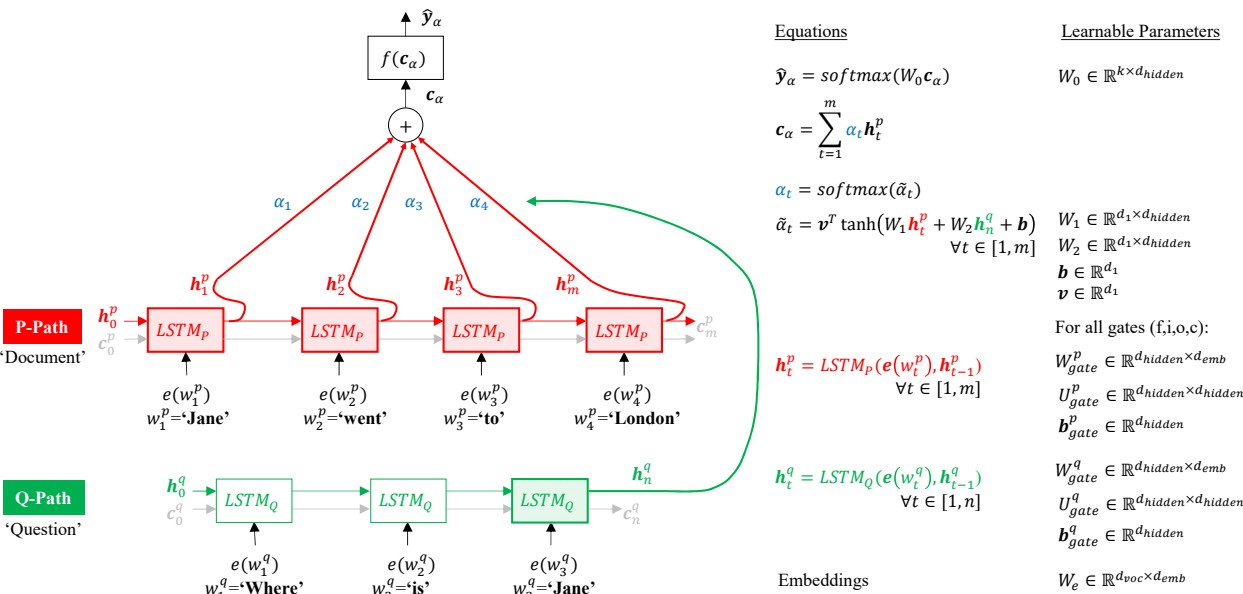

Figure 1: The LSTM+attention model as defined in the paper

**Hyperparameters**   Given the wide variety of tasks and datasets, there is an elaborate set of model- and optimization hyperparameters. Not all parameter values are indicated in the original paper, some were retrieved by inspecting the code (an overview is presented in Appendix A). For all parameters, we used the defaults provided in the original code. We do not engage in further hyperparameter optimization to stay close to the original paper's approach. Note that we are interested in transparency and explainability of the models, not their optimal performance.

**Experimental setup and computational requirements**   We strictly follow the code environment as dictated by the requirements file that accompanies the code. All models are run on Nvdia GTX1080Ti GPUs (11Gb VRAM).

Train and evaluation times varied between datasets and model variations, from ca. 5 minutes (SST dataset) to more than 40 hours (CNN dataset). We ran multiple seeds only on a selection of critical datasets to verify that differences we observed w.r.t. the results in the original paper were significant. Due to resource constraints, all other comparisons are based on single seeding, as was done in the original paper. This means that our observations are indicative, not conclusive.

# 4   Replication of the paper's results

## 4.1   Core replication results

Our reproduction study reveals numerous differences in results reported by Mohankumar et al. (2020), for all datasets where we ran the experiments. Despite the differences, we support the observation that Diversity and Orthogonal LSTM reach similar accuracies as Vanilla LSTM, and lower conicity values, with the same exception reported in the paper (CNN). However, we find the claim that Diversity LSTM leads to more transparent attention distributions is not consistently supported. For Orthogonal LSTM, some results were omitted in the original paper, and we find conflicting results about the effect on faithfulness and plausibility. We present an overview of the comparisons by metric, and the impact our findings have on the main claims of the authors.

104 **Accuracy and conicity**    Of all accuracy and conicity values reported by Mohankumar et al. (2020), we are able
105 to reproduce 86% within a 3%-point margin. Models and datasets that produced the most notable differences are
106 highlighted in Table 1. Despite the different values, the observation that Diversity and Orthogonal LSTM reach similar
107 accuracies as Vanilla LSTM still holds, except for the CNN dataset. Also, we can confirm that conicity values are much
108 lower in Diversity and Orthogonal LSTM, except for CNN in the Diversity LSTM. The largest difference in accuracy
109 we observe for bAbI3, but the output files reveal that the model was not done training after the default 200 epochs.

| Dataset | Accuracy% | | | | | | Conicity | | | | | |
|---|---|---|---|---|---|---|---|---|---|---|---|---|
| | Vanilla LSTM | | Diversity LSTM | | Orthogonal LSTM | | Vanilla LSTM | | Diversity LSTM | | Orthogonal LSTM | |
| | original | rerun | original | rerun | original | rerun | original | rerun | original | rerun | original | rerun |
| SST | 81.79 | 80.3 | 79.95 | 80.0 | 80.05 | 77.6 | 0.68 | 0.71 | 0.20 | 0.19 | 0.28 | 0.28 |
| IMDB | 89.49 | 89.3 | 88.54 | 87.8 | 88.71 | 88.3 | **0.69** | **0.60** | 0.08 | 0.09 | 0.18 | 0.16 |
| Yelp | 95.60 | 94.5 | 95.40 | 93.8 | 96.00 | 94.5 | 0.53 | 0.54 | **0.06** | **0.35** | 0.18 | 0.19 |
| 20News | 93.55 | 90.8 | 91.03 | 90.8 | 92.15 | 91.9 | 0.77 | 0.76 | 0.15 | 0.14 | 0.23 | 0.24 |
| Tweets | **87.02** | **83.3** | 87.04 | 85.4 | 83.20 | 83.9 | 0.77 | 0.78 | 0.24 | 0.23 | 0.27 | 0.26 |
| SNLI | 78.23 | 77.3 | 76.96 | 74.0 | 76.46 | 76.6 | 0.56 | 0.59 | **0.12** | **0.04** | **0.27** | **0.31** |
| QQP | 78.74 | 78.4 | 78.40 | 78.2 | 78.61 | 78.6 | 0.59 | 0.58 | 0.04 | 0.03 | 0.33 | 0.32 |
| bAbI1 | 99.10 | 100.0 | 100.00 | 100.0 | 99.90 | 99.9 | **0.56** | **0.77** | 0.07 | 0.07 | 0.22 | 0.23 |
| bAbI2 | **40.10** | **54.4** | 40.20 | 54.6 | 56.10 | 59.0 | 0.48 | 0.43 | **0.05** | **0.13** | **0.21** | **0.17** |
| bAbI3 | **47.70** | **21.1** | 50.90 | 56.3 | 51.20 | **57.7** | **0.43** | **0.93** | 0.10 | 0.11 | 0.12 | 0.13 |
| CNN | **63.07** | **59.5** | 58.19 | **46.3** | 54.30 | 53.6 | **0.45** | **0.40** | 0.06 | **0.38** | 0.07 | 0.10 |

Table 1: Comparison of reported accuracy and conicity values (differences > 0.03 are highlighted).

110 **Importance of hidden representation**    Mohankumar et al. (2020) analyse the importance of hidden representations
111 using intermediate representation erasure (Serrano and Smith, 2019) and also by examining the effect of permuting the
112 attentions weights (Jain and Wallace, 2019).

113 A visual comparison of the box plots about representation erasure in the paper with box plots in our reruns shows
114 similar results in 25 of the 30 boxes. Despite the fact that our rerun shows lower medians for the box plots for the
115 LSTM in IMDB and 20News dataset, the observation still holds that Diversity LSTM and Orthogonal LSTM reach a
116 quicker decision flip for SS tasks. We concur with the authors' observations on the paraphrase detection (QQP) and
117 Q&A task (bAbI1). In our rerun we see that the quick decision flip that is shown in bAbI1 also occurs in bAbI2 and
118 bAbI3. Mohankumar et al. (2020) do not report on SNLI and CNN, where our rerun shows no improvement of the
119 Diversity LSTM and Orthogonal LSTM models over Vanilla LSTM.

120 The impact of permuting attention weights is difficult to compare with our results as Mohankumar et al. (2020) only
121 report a graphical representation (violin plots) of median output difference. After visual comparison we judge that the
122 overall results are similar for IMDB, 20News and Yelp. We also evaluate the median output difference for datasets not
123 reported by Mohankumar et al. (2020). We observe that the results for SST and Tweets show a similar 'shift to the right'
124 as reported for other binary classification tasks. For DS tasks we observe that Vanilla LSTM already has relatively high
125 median output difference, and the Diversity LSTM and Orthogonal LSTM provide less improvement.

126 We conclude that in our experiments, the Diversity and Orthogonal LSTM do result in quicker decision flips and higher
127 output difference for SS tasks, but not consistently for the other tasks.

128 **Comparison with rationales**    Our rerun of rationale length and rationale attention shows very different results as
129 reported by Mohankumar et al. (2020), see Table 2. Although we can confirm that Diversity LSTM results in shorter
130 rationales, we cannot support the claim that Diversity LSTM provides much higher attention to the rationale than
131 Vanilla LSTM. In our rerun this only holds for 20News.

132 The data for the Orthogonal LSTM, which were not reported by Mohankumar et al. (2020), show much shorter rationale
133 length, consistent with the paper's claim. However, impact on the share of attention on the rationale is mixed: it is
134 higher for Yelp and 20 News, similar for IMDB and Tweets, but lower for SST.

135 For DS tasks, the rationale comparison is not implemented by the authors, we suspect because of the high computational
136 costs involved for calculating rationales in tasks with multiple output categories.

137 **Comparison with attribution methods**    The rerun of the correlation metrics shows numerous differences in both
138 Pearson correlation and JS Divergence. After studying Pearson correlation, we support the authors' claim that compared
139 with Vanilla LSTM, Diversity LSTM produces attention weights that better correlate with gradients and integrated
140 gradients, although in our results the relative increase of correlation with gradients is smaller: 13% instead of the 65%
141 reported by Mohankumar et al. (2020)[7]. However, we do not see the claimed reduction in JS Divergence. In fact, for

---

[7]This percentage represents the average of the increases over all datasets.

| Dataset | Rationale attention | | | | | | Rationale length | | | | | |
|---|---|---|---|---|---|---|---|---|---|---|---|---|
| | Vanilla LSTM | | Diversity LSTM | | Orthogonal LSTM | | Vanilla LSTM | | Diversity LSTM | | Orthogonal LSTM | |
| | original | rerun | original | rerun | original | rerun | original | rerun | original | rerun | original | rerun |
| SST | **0.348** | **0.74** | 0.624 | 0.55 | - | 0.35 | **0.240** | **0.72** | 0.175 | 0.18 | - | 0.10 |
| IMBD | **0.472** | **0.97** | **0.761** | **0.91** | - | 0.92 | **0.217** | **0.92** | **0.169** | **0.22** | - | 0.27 |
| Yelp | 0.438 | 0.43 | **0.574** | **0.27** | - | 0.55 | **0.173** | **0.38** | 0.160 | 0.19 | - | 0.11 |
| 20News | 0.627 | 0.62 | 0.884 | 0.94 | - | 0.86 | **0.215** | **0.59** | **0.173** | **0.27** | - | 0.24 |
| Tweets | **0.284** | **0.82** | **0.764** | **0.59** | - | 0.79 | **0.225** | **0.81** | 0.306 | 0.32 | - | 0.39 |

Table 2: Comparison of reported rationales (differences > 0.05 are highlighted)

all datasets the Diversity LSTM produces similar or even higher JS Divergence values than Vanilla LSTM, except JS Divergence with Integrated Gradients for 20News, see Table 3. The Orthogonal LSTM, for which no correlation data is reported in the paper, is in line with the Diversity LSTM in this respect.

| Dataset | JS Divergence Gradients | | | | | | JS Divergence Integrated Gradients | | | | | |
|---|---|---|---|---|---|---|---|---|---|---|---|---|
| | Vanilla LSTM | | Diversity LSTM | | Orthogonal LSTM | | Vanilla LSTM | | Diversity LSTM | | Orthogonal LSTM | |
| | original | rerun | original | rerun | original | rerun | original | rerun | original | rerun | original | rerun |
| SST | 0.10 | 0.09 | 0.08 | 0.09 | - | 0.14 | 0.12 | 0.10 | 0.09 | 0.10 | - | 0.15 |
| IMDB | 0.09 | 0.08 | 0.09 | 0.11 | - | 0.13 | 0.13 | 0.11 | 0.13 | 0.15 | - | 0.18 |
| Yelp | 0.15 | 0.12 | **0.13** | **0.17** | - | 0.16 | 0.19 | 0.18 | 0.19 | 0.19 | - | 0.17 |
| 20News | 0.15 | 0.18 | **0.06** | **0.17** | - | 0.17 | 0.21 | 0.22 | **0.07** | **0.13** | - | 0.15 |
| Tweets | 0.08 | 0.07 | 0.12 | 0.09 | - | 0.18 | 0.08 | 0.08 | **0.15** | **0.10** | - | 0.19 |
| SNLI | 0.11 | 0.11 | 0.10 | 0.11 | - | 0.12 | 0.16 | 0.14 | 0.13 | 0.14 | - | 0.15 |
| QQP | **0.15** | **0.10** | 0.10 | 0.11 | - | 0.12 | **0.19** | **0.15** | 0.15 | 0.14 | - | 0.14 |
| bAbI1 | **0.33** | **0.12** | 0.21 | 0.23 | - | 0.21 | **0.43** | **0.25** | 0.24 | 0.22 | - | 0.28 |
| bAbI2 | **0.53** | **0.39** | **0.23** | **0.40** | - | 0.38 | **0.58** | **0.51** | **0.19** | **0.58** | - | 0.54 |
| bAbI3 | **0.46** | **0.26** | 0.37 | 0.36 | - | 0.43 | **0.64** | **0.35** | **0.41** | **0.64** | - | 0.64 |
| CNN | **0.22** | **0.16** | **0.17** | **0.34** | - | 0.39 | **0.30** | **0.23** | **0.21** | **0.51** | - | 0.44 |

Table 3: Comparison of correlation metrics. Differences > 0.03 are highlighted

**Analysis by POS tags**    A comparison of the importance that is attributed to various POS tags shows similar importance and ranking for the SST, 20News and Tweets datasets. For Yelp and QQP we get different outcomes. Most notably, with vanilla LSTM model for Yelp we see no attention given to punctuations (PUNC), for which Mohankumar et al. (2020) reports highest attention. For QQP, Mohankumar et al. (2020) reports 23% on PUNC, while we find only 9%. Our results indicate the improvements shown in POS tags are less clear than reported by Mohankumar et al. (2020).

**Human evaluation**    We could not reproduce the human evaluation within the four-week time frame of our research. Mohankumar et al. (2020) reports convincing results, and we also believe human interpretation should play a key role in judging whether their methods improve transparency. We include some examples in Appendix D for this purpose.

### 4.2   Conclusion regarding reproducibility

Our findings are summarized in Table 4. We conclude that it is not immediately clear that Diversity LSTM and Orthogonal LSTM provide better transparency for all the studied datasets.

- The Orthogonal LSTM clearly leads to lower conicity than Vanilla LSTM, but Mohankumar et al. (2020) show little evidence with other metrics that indicate higher faithfulness: of the 14 datasets, only 6 boxplots and 4 violin charts are included. The results observed in our rerun are mixed. For example, Orthogonal LSTM works well for 20News, but for SNLI there is hardly any effect on the box plot, and also correlation/JSD with (integrated) gradients is worse.

- For Diversity LSTM, Mohankumar et al. (2020) show convincing evidence with substantial data. We observe similar trends in conicity, and the impact of diversity training is clear in the box plots and violin charts for the binary classification tasks. However, for the tasks that require two input sequences like SNLI, bAbI2, CNN our rerun shows that Diversity LSTM does not contribute much to faithfulness and can lead to lower correlation with (integrated) gradients and higher JS Divergence.

## 5   Code Review

As part of the reproduction study, we familiarized ourselves with the code to understand how the model and the experiments had been implemented. We also scrutinized the code to check whether we could find a cause for the differences we found in the reported metrics. The code's class architecture can accommodate a wide range of tasks,

| Metric | Claim (with reference to paragraph number in Mohankumar et al. (2020)) | Supported after rerun | Notes |
|---|---|---|---|
| Accuracy and conicity | Diversity LSTM and Orthogonal LSTM achieve similar accuracies as Vanilla LSTM, but much lower conicity (§5.2) | Yes | Except CNN in Diversity LSTM |
| Fraction of hidden representation required for decision flip (box plots) | Diversity LSTM and Orthogonal LSTM reach quicker decision flip (§5.3) | Yes (for models in paper) | Especially for BC tasks, somewhat for QQP; not for SNLI and QA tasks |
| Median Output difference on randomly permuting attention weights (violin charts) | Diversity LSTM and Orthogonal LSTM are more sensitive to random permutation of weights than Vanilla LSTM (§5.3) | Yes (for models in paper) | Clear difference for BC tasks, mixed picture for the dual-sequence tasks. |
| Rationale attention | Diversity LSTM provides much higher attention to rationales than Vanilla LSTM across the 8 Text classification datasets (§5.4) | No | Only true for 20News; No results reported on QA tasks |
| Rationale length | Diversity LSTM often provides shorter rationales than Vanilla LSTM (§5.4) | Yes | No results reported on QA tasks |
| Pearson correlation and JS divergence between distribution of attention and (integrated) gradients | Attention weights in Diversity LSTM better agree with gradients and integrated gradients than Vanilla LSTM (§5.5) | Mixed | Diversity LSTM has higher Pearson correlation, but similar or higher JS Divergence |
| Attention given to POS tags | Attention given to punctuation marks is significantly reduced on the Yelp, Amazon and QQP datasets (§5.6) | No | Not for Yelp, less clear for QQP |
|  | Diversity LSTM gives much more attention to adjectives than Vanilla LSTM in the four sentiment analysis tasks (SST, IMDB, Yelp, Amazon) (§5.6) | Yes | True for SST and IMDB, but not for Yelp |
| Human evaluation of plausibility | Human evaluators prefer attention distribution of Diversity LSTM over Vanilla LSTM for Yelp, SNLI, QQP and bAbI1 (§5.7) | Not reproduced | Evaluation by only 15 people |

Table 4: Evidence for authors' claims after rerun

datasets and model configurations. While convenient, this also makes the codebase complex and susceptible to errors. The code review revealed several debatable choices, of which the main ones are described below.

**Orthogonalization of Q-path in dual input sequence tasks** For DS tasks, we expect the orthogonalization procedure to only be activated in the P-path (the path of the input document) of the model, as this is the path on which the attention mechanism applies its weights $\alpha_t$. However, in the code, orthogonalization is *also* applied to the Q-path (the path of the question phrase in the Q&A tasks, or the second input phrase in SNLI and QQP).

In our view, this introduces a potentially problematic effect. The attention mechanism uses only the last hidden state vector $\mathbf{h}_t^q$ as the query term. This representation for the last word in the sequence will only retain the vector component orthogonal to the mean of the previous word representations, as a result of orthogonalization. We argue that the direction of $\mathbf{h}_t^q$ in the hidden space will represent the exclusive 'change of meaning' that the last word adds to the sequence. This is not a problem in the bAbI tasks, where the prompt word in the question phrase is always the last word (e.g., 'Where is Jane'). But for longer questions where the prompt words appear earlier in the question, this may impede the attention mechanism from finding the right prompt words.

In order to test this sensitivity, we conduct an experiment for the simpler SS tasks. We apply orthogonalization during training and compare model performance when i) attention weights are left unconstrained vs. ii) all attention weights are set to zero, except for the last hidden state. The result is shown in Table 5. What is striking is the performance remains on par (marked in green) for Vanilla LSTM when only attending to the last hidden state, indicating the model performs well without the attention mechanism. However, we observe a performance drop of 10%-34% (absolute) when attention is constrained for the Orthogonal LSTM (marked in red). Indeed, it appears part of the information required for inference is lost.

How this effect impacts the results requires further study. It may explain the accuracy drop from 63% (Vanilla LSTM) to 58%/54%(Diversity/Orthogonal LSTM) for CNN as reported in Table 2 by Mohankumar et al. (2020). We have contacted the authors to verify their intentions, but did not receive a response prior to submission of this reproduction study.

| | Vanilla LSTM | | Orthogonal LSTM | |
|---|---|---|---|---|
| Dataset | Base attention | last_only attention | Base attention | last_only attention |
| SST: accuracy | 0.803 | **0.810** | 0.776 | **0.583** |
| (conicity) | (0.713) | (0.763) | (0.283) | (0.265) |
| IMDB: accuracy | 0.893 | **0.876** | 0.883 | **0.784** |
| (conicity) | (0.602) | (0.885) | (0.163) | (.141) |
| 20News: accuracy | 0.908 | **0.857** | 0.919 | **0.583** |
| (conicity) | (0.761) | (0.831) | (0.235) | (0.395) |
| Tweets: accuracy | 0.833 | **0.782** | 0.839 | **0.712** |
| (conicity) | (0.776) | (0.798) | (0.260) | (0.330) |

Table 5: Demonstration of adverse effect of orthogonalization on the information content of the last hidden state vector (results reflect our experiments, not the original paper)

**Disparate calculation of final prediction** For DS tasks, in the code the final prediction layer is implemented as $\hat{y} = \mathrm{softmax}(\mathbf{W}_r(\tanh(\mathbf{W}_p\mathbf{c}_\alpha + \mathbf{b}_p + \mathbf{W}_q\mathbf{h}_n^q + \mathbf{b}_q)) + \mathbf{b}_r)$. This deviates from the prediction function $\hat{y} = \mathrm{softmax}(\mathbf{W}_0\mathbf{c}_\alpha)$

202 described in Section 2.1 by Mohankumar et al. (2020)[8]. However, this does not affect the core architecture, namely
203 LSTM and attention, so we did not modify the code or conduct further experiments.

204 **Fine-tuning of embeddings**   The models use pre-trained embeddings except for the bAbI datasets. Words outside of
205 the pre-trained embeddings' vocabulary are initialized with zero-vectors. All embeddings are fine-tuned (i.e. trainable),
206 independently for the P- and Q-paths for DS tasks. This is not mentioned in the original paper and this choice is
207 questionable as it leads to an excessive number of trainable parameters (e.g., >40M for the CNN dataset, see Appendix
208 A) and training time, while it is unlikely to be critical for the tasks.

209 **Definition of dev set for bAbI datasets**   While pre-processing bAbI datasets, 15% of the train set is randomly selected
210 to be used as dev set, resulting in much higher similarity between these two splits compared to the test set. As a result,
211 the trained model is overfit on the train set, and we observe a large gap between dev and test accuracy.

## 6   Extension of the evaluation methods

213 As discussed in Section 4.1, our rerun of Pearson's correlation and JS Divergence between attention weights and
214 gradients/integrated gradients points towards a less convincing conclusion. We therefore also used the LIME framework
215 (Ribeiro et al., 2016) as a third metric for comparing how transparent the attention weights are as explanations, as well
216 as how much improvements are brought about by the Diversity and Orthogonal LSTM.

217 We use LIME to generate a score for the predicted class on each
218 word-position in the sentence, which can then be compared
219 with the attention weights. For calculating JS divergence we
220 also rescaled the lime score so that the scores range from 0 to 1,
221 and sums to 1 per sentence (i.e. similar to attention scores). The
222 results are shown in Table 6, where we experimented with only
223 a representative selection of datasets due to time and resource
224 constraints.

| Dataset | Pearson's Correlation | | | JS Divergence | | |
|---|---|---|---|---|---|---|
| | Vanilla | Ortho. | Div. | Vanilla | Ortho. | Div. |
| IMDB | 0.42 | **0.33** | 0.42 | 0.26 | **0.44** | **0.42** |
| 20News | 0.30 | **0.70** | **0.71** | 0.22 | **0.42** | **0.45** |
| Tweets | 0.13 | **0.38** | **0.43** | 0.07 | **0.33** | **0.18** |
| SNLI | 0.24 | 0.22 | 0.23 | 0.15 | 0.15 | 0.12 |
| bAbI1 | 0.69 | 0.67 | **0.58** | 0.42 | 0.38 | 0.46 |

Numbers that agree with expectations (higher correlation, lower JS Divergence) are highlighted in **green**, numbers opposite to expectations are highlighted in **red**.

Table 6: Correlation and JS Divergence between attention weights and LIME scores

225 Similar to our comparison of attention weights with gradient-
226 based methods, Table 6 indicates Diversity and Orthogonal
227 LSTM fail to produce explanations consistent with LIME. It is
228 also not clear which statistical measure is best for comparing
229 whether two explanation methods agree with each other. In
230 several instances (e.g. 20News and Tweets), we observe an increase in Pearson's correlation and an increase in JS
231 Divergence at the same time when going from Vanilla LSTM to Orthogonal/Diversity LSTM models.

## 7   Generalization to other model architectures

233 Despite the differences we found between our observations and the observations reported by (Mohankumar et al., 2020),
234 we still see the potential value of the methods they propose. This is because we did observe sparser attention weights
235 when using Diversity and Orthogonal LSTM, and because of the strong preference expressed for the Diversity LSTM in
236 the human evaluations conducted by Mohankumar et al. (2020).

237 We therefore investigate how well these methods work in alternative settings. So far the Orthogonalization and Diversity
238 Training methods are only tested on one-layer uni-directional LSTM models with attention. However, in many recent
239 studies, BiLSTM-based attention models or Transformer models are used (Zhou and Wu, 2018; Lee et al., 2019; Aziz
240 Sharfuddin et al., 2018). Similarly, more complex attention mechanisms such as self-attention and multi-head attention
241 (Vaswani et al., 2017) gained popularity due to their superior performance. For this reason, we investigate whether the
242 proposed methods can be extended to more complex models and whether the authors' two main claims still apply.

243 **Extending to other attention mechanisms**   The application of more advanced attention mechanisms (such as multi-
244 head attention) poses a challenge because they produce more than one attention weight per word. It is not straightforward
245 to generate explanations and word importance based on these weights. As a consequence, several of the evaluation
246 metrics used by the authors cannot be applied in their current form. This would require making non-trivial design
247 choices on how to combine multiple distributions of the attention weights. Further research is required to investigate this
248 and whether existing methods such as Attention Flow and Attention Rollout (Abnar and Zuidema, 2020) can provide a
249 resolution.

---

[8] `https://github.com/akashkm99/Interpretable-Attention/blob/master/model/modules/Decoder.py#L101-L107`

**Extending to other architectures: BiLSTM Experiments**   We replace the uni-directional LSTM in the model (Figure 1) with a bi-directional LSTM. We choose the BiLSTM architecture, and not a Transformer based architecture, as the latter requires dealing with the more advanced attention mechanisms discussed above.

In order to maintain the decoder's complexity (the attention mechanism), we preserve the output dimension of the LSTM. This requires halving the dimension of the hidden states, which also ensures that the number of trainable weights of the BiLSTM is comparable to that of the unidirectional LSTM. For the Diversity BiLSTM, the same diversity weights are used as in Mohankumar et al. (2020). The conicity term present in the loss function of the Diversity BiLSTM is calculated based on the concatenated forward and backward hidden representations. Orthogonalization for the Ortho BiLSTM is applied before concatenation of the forward and backward hidden states.

Results show that the application of the two methods proposed by Mohankumar et al. (2020) do not result in performance loss and do lower conicity. However, on other metrics and across datasets, the picture is mixed like we saw in our reproducibility results for the unidirectional LSTM, indicating the methods do not unconditionally improve explanations. We will not discuss these results in detail, but conclude that it is indeed possible to extend the proposed methods to BiLSTM attention models. Full results are included in Appendix C for completeness.

# 8   Discussion

Our reproduction shows that enforcing low conicity between the hidden states of an LSTM encoder does not guarantee improved transparency in the studied datasets, at least not on the metrics used by Mohankumar et al. (2020). We find the authors' claim about improved transparency not generally applicable and under certain conditions their methods even hurt accuracy. Still, the Diversity LSTM and Orthogonal LSTM do lead to improved metrics on some datasets, and the human evaluation Mohankumar et al. (2020) conducted shows strong preference for the Diversity LSTM over Vanilla LSTM. This raises the question under what conditions these methods should be applied.

**Conditions underlying effectiveness**   One pattern that seems to emerge is that the benefits of orthogonalizing or diversity training are most apparent for the relatively simpler SS tasks. The potential to improve faithfulness of the weights might be high in those cases as it not a given that attention weights carry any meaning for these task.

For some tasks, the LSTM does not strictly need the attention mechanism to perform well, as is shown in Table 7 when the attention mechanism is constrained to be either uniform or attending to the last word only. In contrast, the more difficult DS tasks do require the attention mechanism in order to reach higher accuracies. This pattern is similar to that described by Wiegreffe and Pinter (2019).

| Dataset | Base attention | | Constrained attention | |
|---|---|---|---|---|
| | Reported | Repr. | uniform | last_only |
| SST | .818 | .803 | **.800** | **.810** |
| IMDB | .895 | .893 | **.883** | **.876** |
| Yelp | .956 | .949 | **.950** | **.949** |
| 20News | .936 | .908 | **.898** | **.857** |
| Tweets | .870 | .833 | **.833** | **.782** |
| SNLI | .782 | .773 | **.755** | **.759** |
| QQP | .787 | .784 | **.789** | **.792** |
| bAbI1 | .991 | 1.00 | **.485** | **.729** |
| bAbI2 | .401 | .544 | **.315** | **.441** |
| bAbI3 | .477 | .211* | - | - |
| CNN | .631 | .595 | **.424** | **.367** |

\* Reproduction failed, comparisons not applicable

Table 7: Impact on performance of the Vanilla LSTM when forcing uniform, first- and last only attention

We suspect that there is a relation between a) how crucial the attention mechanism is for performance in a given task, b) how much improvement Orthogonal/Diversity LSTM can offer w.r.t. plausibility of the attention weights for explaining the model's outputs. This relationship, and the conditions under which orthogonalization and diversity training offer the best results, deserves additional investigation.

**Reflection on our replication study**   A key insight we have gained is that even with access to the original code, exact reproduction of the results is not guaranteed. We have not been able to find the cause of several differences in results. The available time and hardware limited our possibilities to repeat these experiments with multiple seeds to find an estimate of the variance of outcomes.

Another insight we gained is that the metrics concerning faithfulness and plausibility can be hard to interpret, as it is deeply entangled with the nature of the dataset as well as the model implementation. To enable scalable development of transparent AI models, reliable quantitative metrics are needed that can accurately approximate real humans' judgement. We believe further development of transparency metrics is an important area for further research to help build more transparent models.

# Acknowledgement

We'd like to thank [name to be added] for his guidance and insightful discussions.

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

# Appendix A: Details of models and datasets

| | Single input sequence tasks | | | | | | Dual input sequence tasks | | | | | |
|---|---|---|---|---|---|---|---|---|---|---|---|---|
| | **SST** | **IMDB** | **Yelp** | **Amazon** | **20News** | **Tweets** | **SNLI** | **QQP** | **Babi1** | **Babi2** | **Babi3** | **CNN** |
| _Model configuration_ | | | | | | | | | | | | |
| Model LSTM variation | Vanilla | Vanilla | Vanilla | Vanilla | Vanilla | Vanilla | Vanilla | Vanilla | Vanilla | Vanilla | Vanilla | Vanilla |
| Attention type | tanh | tanh | tanh | tanh | tanh | tanh | tanh | tanh | tanh | tanh | tanh | tanh |
| Embedding dim | 300 | 300 | 300 | 300 | 300 | 300 | 300 | 300 | 50 | 50 | 50 | 300 |
| Embedding voc | 13.826 | 12.487 | 63.328 | 49.883 | 6.515 | 6.845 | 20.981 | 26.635 | 24 | 38 | 39 | 70.190 |
| Pre-embed | FastText | FastText | FastText | FastText | FastText | FastText | GLOVE | GLOVE | None | None | None | FastText |
| LSTM hidden dim | 256 | 256 | 256 | 256 | 256 | 256 | 256 | 256 | 64 | 128 | 128 | 256 |
| Output size | 1 | 1 | 1 | 1 | 1 | 1 | 3 | 2 | 36 (6)* | 36 (6)* | 36 (6)* | 584 (26)* |
| _Optimizer hyperparameters_ | | | | | | | | | | | | |
| Diversity weight (if applic | 0.5 | 0.5 | 0.5 | 0.5 | 0.5 | 0.5 | 0.1 | 0.5 | 0.5 | 0.5 | 0.5 | 0.2 |
| Batch size | 32 | 32 | 32 | 32 | 32 | 32 | 128 | 128 | 32 | 64 | 64 | 90 |
| Optimizer | Adam | Adam | Adam | Adam | Adam | Adam | Adam | Adam | Adam | Adam | Adam | Adam |
| LR | 0.001 | 0.001 | 0.001 | 0.001 | 0.001 | 0.001 | 0.001 | 0.001 | 0.001 | 0.001 | 0.001 | 0.001 |
| Weight decay | 1,E-05 | 1,E-05 | 1,E-05 | 1,E-05 | 1,E-05 | 1,E-05 | 1,E-05 | 1,E-05 | 1,E-05 | 1,E-05 | 1,E-05 | 1,E-05 |
| Epochs | 8 | 8 | 8 | 8 | 8 | 8 | 25 | 25 | 100 | 200 | 200 | 12 |
| _Trainable weights, including fine-tuning of embeddings_ | | | | | | | | | | | | |
| Pencoder | 4.719.192 | 4.317.492 | 19.569.792 | 15.536.292 | 2.525.892 | 2.624.892 | 6.865.692 | 8.480.892 | 30.896 | 94.060 | 94.110 | 21.628.392 |
| Qencoder | 0 | 0 | 0 | 0 | 0 | 0 | 6.865.692 | 8.480.892 | 30.896 | 94.060 | 94.110 | 21.628.392 |
| Decoder | 33.281 | 33.281 | 33.281 | 33.281 | 33.281 | 33.281 | 132.099 | 131.970 | 9.540 | 35.428 | 35.428 | 207.048 |
| Total | 4.752.473 | 4.350.773 | 19.603.073 | 15.569.573 | 2.559.173 | 2.658.173 | 13.863.483 | 17.093.754 | 71.332 | 223.548 | 223.648 | 43.463.832 |
| _Trainable weights, without fine-tuning of embeddings_ | | | | | | | | | | | | |
| Pencoder | 571.392 | 571.392 | 571.392 | 571.392 | 571.392 | 571.392 | 571.392 | 571.392 | n/a | n/a | n/a | 571.392 |
| Qencoder | 0 | 0 | 0 | 0 | 0 | 0 | 571.392 | 571.392 | n/a | n/a | n/a | 571.392 |
| Decoder | 33.281 | 33.281 | 33.281 | 33.281 | 33.281 | 33.281 | 132.099 | 131.970 | n/a | n/a | n/a | 207.048 |
| Total | 604.673 | 604.673 | 604.673 | 604.673 | 604.673 | 604.673 | 1.274.883 | 1.274.754 | n/a | n/a | n/a | 1.349.832 |

* Output size is numer of total entities in the dataset, part of which is masked in each datapoint (numer of categories used on average per data point)

Table 8: Model- and hyperparameters for standard configurations per dataset

| Dataset | Description | Number of datapoints | | | Avg seq. length (train) | | Avg.no.answer | Vocab. size |
|---|---|---|---|---|---|---|---|---|
| | | train (%pos) | dev (%pos) | test (%pos) | Document | Question | categories | (train, docs) |
| **Single input sequence tasks** | | | | | | | | |
| SST | Sentiment analysis | 6,355 (52%) | 821 (52%) | 1,725 (50%) | 21 | n/a | 2 | 13,703 |
| IMDB | Sentiment analysis | 17,200 (50%) | 4,297 (49%) | 4,356 (50%) | 182 | n/a | 2 | 12,486 |
| Yelp | Sentiment analysis | 345,285 (54%) | 4,790 (54%) | 26,866 (54%) | 74 | n/a | 2 | 63,304 |
| Amazon* | Sentiment analysis | 1,528,080 (52%) | 4,456 (52%) | 331,774 (52%) | 57 | n/a | 2 | 49,881 |
| Anemia* | Diagnosis prediction | - | - | - | - | - | - | - |
| Diabetes* | Diagnosis prediction | - | - | - | - | - | - | - |
| 20News | Topic classification | 1,145 (50%) | 278 (50%) | 357 (50%) | 119 | n/a | 2 | 5,904 |
| Tweets | Topic classification | 13,938 (12%) | 2,447 (13%) | 4,123 (12%) | 23 | n/a | 2 | 6,841 |
| **Dual input sequence tasks** | | | | | | | | |
| SNLI | Natural language inference | 549,367 | 9,842 | 9,824 | 16 | 10 | 3 | 17,943 |
| QQP | Paraphrase detection | 327,460 (37%) | 36,384 (37%) | 40,430 (37%) | 15 | 15 | 2 | 26,172 |
| bAbI1 | Question answering | 8,500 | 1,500 | 1,000 | 38 | 5 | 6 | 20 |
| bAbI2 | Question answering | 8,500 | 1,500 | 1,000 | 96 | 6 | 6 | 34 |
| bAbI3 | Question answering | 8,500 | 1,500 | 1,000 | 309 | 9 | 6 | 34 |
| CNN | Question answering | 380,298 | 3,924 | 3,198 | 764 | 14 | 26.1 | >70,000 |

* Replication could not be performed for these datasets due to either availability or memory size limits

Table 9: Characteristics of the datasets

# Appendix B: Full replication results

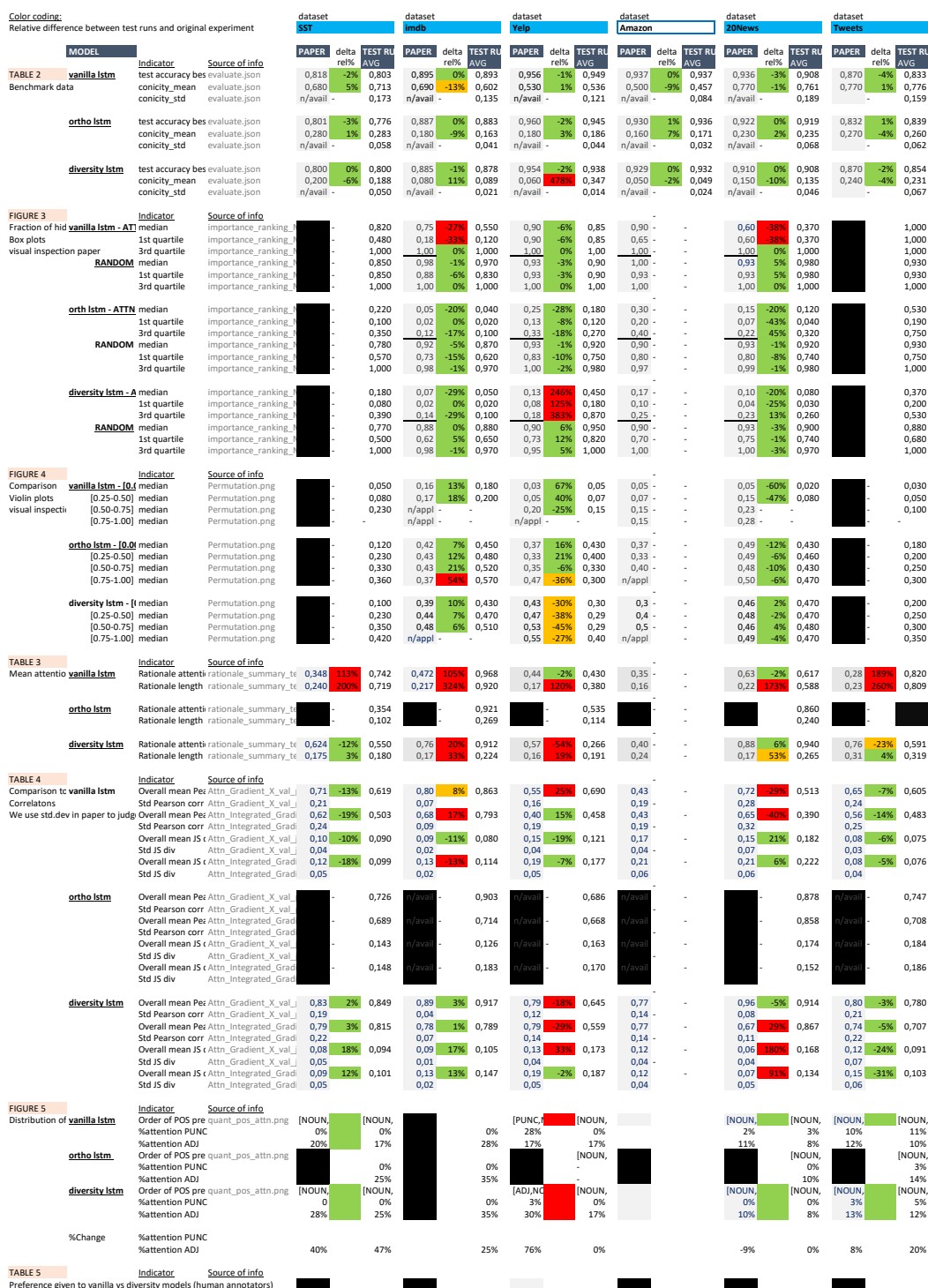

Figure 2: Replication results for single sequence tasks

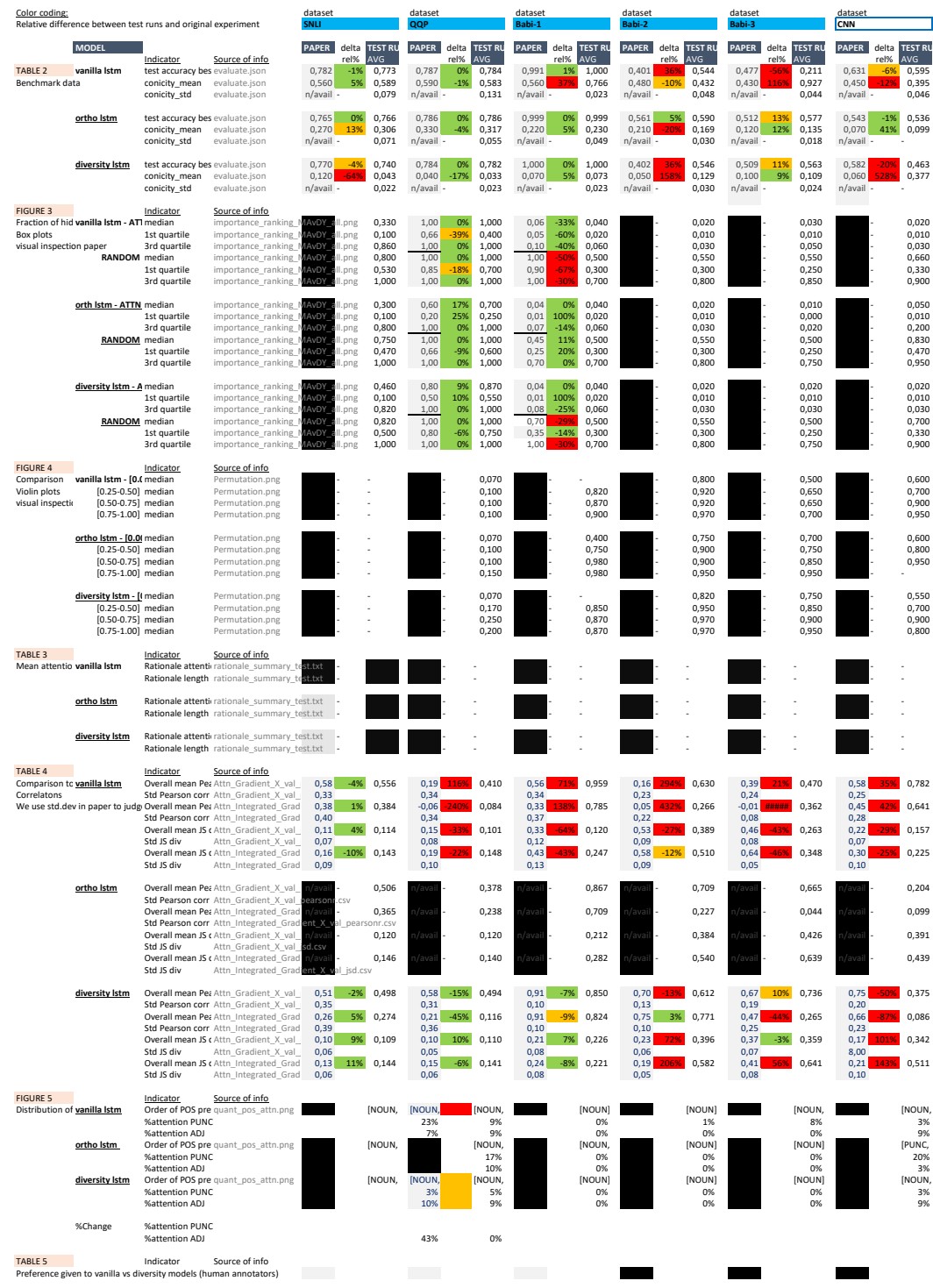

Figure 3: Replication results for dual sequence tasks

# Appendix C: Results of BiLSTM extension

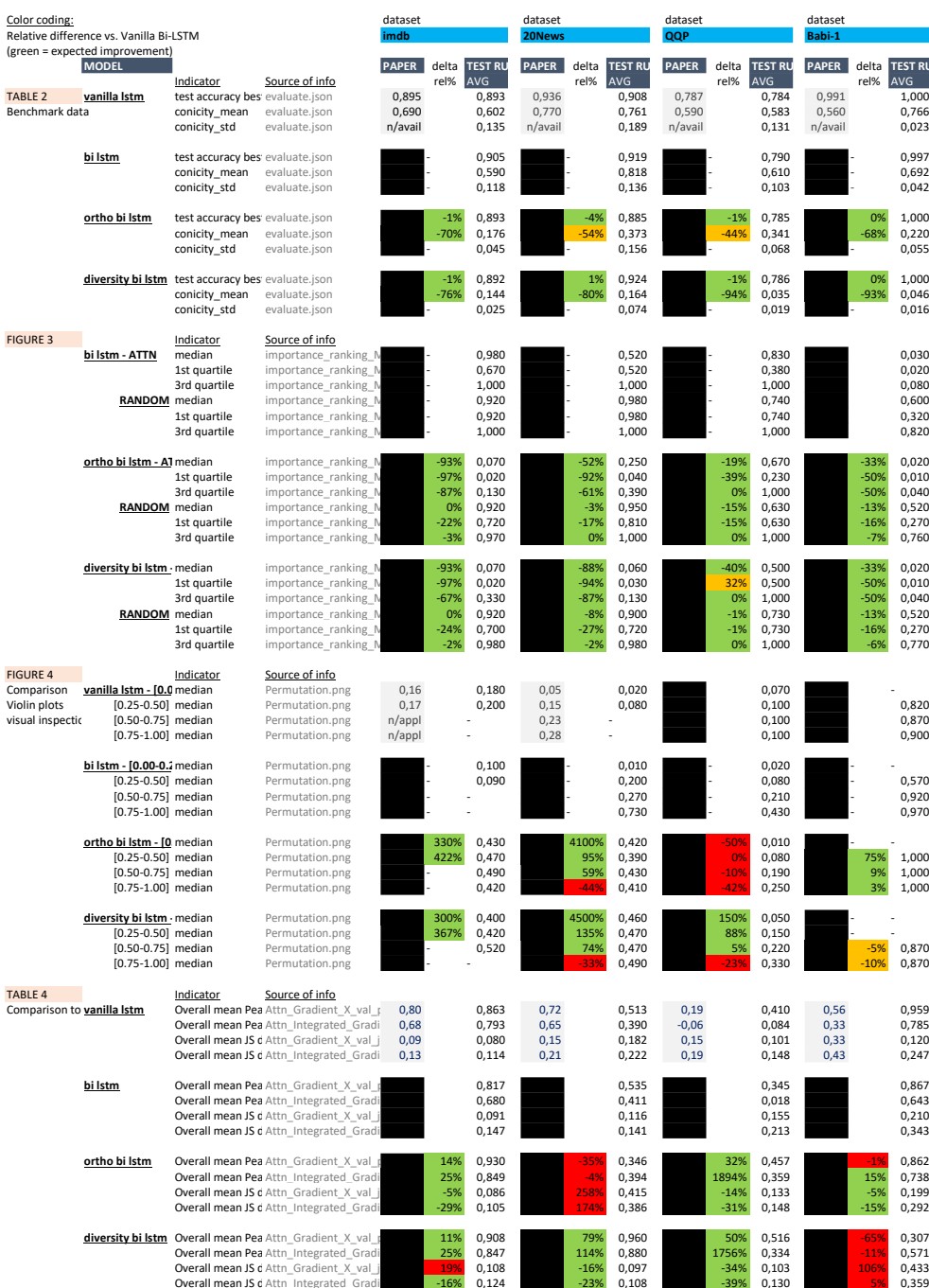

Figure 4: Performance of BiLSTM on evaluation metrics

# Appendix D: Selected data examples and illustration of model behavior

## SST examples

```
SST, Vanilla-LSTM, trained on default (tanh) attention (test acc=.805, conicity=.697)

Sentence  : <SOS>    a    slick    skillful    little    horror    film    <EOS>
Attentions : 0.00    0.04  0.08     0.20        0.30      0.24      0.13    0.00
SM(||h_i||): 0.00    0.09  0.17     0.24        0.17      0.17      0.17    0.00
Label:  1 , Prediction: [0.5812621]

SST, Ortho-LSTM, trained on default (tanh) attention (test acc=.776, conicity=.283)

Sentence  : <SOS>    a    slick    skillful    little    horror    film    <EOS>
Attentions : 0.00    0.00  0.03     0.64        0.18      0.04      0.11    0.00
SM(||h_i||): 0.00    0.15  0.14     0.21        0.10      0.16      0.25    0.00
Label:  1 , Prediction: [0.7968075]

SST, Diversity-LSTM, trained on default (tanh) attention (test acc=.800, conicity=.188)

Sentence  : <SOS>    a    slick    skillful    little    horror    film    <EOS>
Attentions : 0.00    0.00  0.12     0.86        0.01      0.01      0.01    0.00
SM(||h_i||): 0.00    0.02  0.04     0.19        0.25      0.16      0.34    0.00
Label:  1 , Prediction: [0.9623399]
```

## Yelp examples

```
Yelp, Vanilla-LSTM, trained on default (tanh) attention (test acc=.949, conicity=.536)

Sentence  : <SOS>   Been   going   here    for    years.    A    great    place!   <EOS>
Attentions : 0.00   0.35    0.09   0.06   0.01    0.01     0.04  0.18     0.26     0.00
SM(||h_i||): 0.00   0.08    0.07   0.10   0.08    0.09     0.06  0.18     0.35     0.00
Label:  1 , Prediction: [0.984035]

Yelp, Ortho-LSTM, trained on default (tanh) attention (test acc=.945, conicity=.186)

Sentence  : <SOS>   Been   going   here    for    years.    A    great    place!   <EOS>
Attentions : 0.00   0.53    0.00   0.00   0.00    0.08     0.00  0.22     0.17     0.00
SM(||h_i||): 0.00   0.18    0.08   0.05   0.07    0.04     0.03  0.34     0.21     0.00
Label:  1 , Prediction: [0.99045765]

Yelp, Diversity-LSTM, trained on default (tanh) attention (test acc=.938, conicity=.347)

Sentence  : <SOS>   Been   going   here    for    years.    A    great    place!   <EOS>
Attentions : 0.00   0.41    0.16   0.00   0.00    0.02     0.09  0.21     0.10     0.00
SM(||h_i||): 0.00   0.18    0.05   0.02   0.04    0.03     0.06  0.40     0.21     0.00
Label:  1 , Prediction: [0.9969946]
```

Figure 5: Examples of single input sequence tasks

## CNN examples

CNN, Vanilla-LSTM, trained with default (tanh) attention (test acc=.595, conicity=.395)

```
P path      : <SOS>      (      @entity2      )      one  @entity1  citizen  was      killed  and      another  injured  in      what
Attentions : 0.00       0.00  0.00          0.00  0.04  0.06      0.08     0.01      0.00    0.00     0.00     0.00    0.00    0.00
police  are      calling  a      suspected  terror  attack  wednesday  night      near      @entity6  .      @entity8
0.00    0.00     0.00     0.00   0.00       0.00    0.00    0.00       0.00       0.00      0.00      0.00   0.07
spokesman  @entity7  said  a  37  -  year  -  old  @entity10  motorist  from  @entity11  struck  two
0.02       0.00      0.00     0.00  0.00  0.00  0.00  0.00  0.43       0.00      0.00  0.03       0.00    0.00
people  standing  at  a  bus  stop  in  the  @entity15  section  of  the  city  .  one  victim  ,
0.00    0.00      0.00  0.00  0.00  0.00  0.00  0.00  0.00       0.00     0.00  0.00  0.00  0.00  0.00  0.00
identified  by  police  as  @entity18  ,  26  ,  died  at  the  hospital  .  a  20  -  year  -
0.00        0.00  0.00  0.00  0.20       0.00  0.00  0.00  0.00  0.00  0.00  0.00       0.00  0.00  0.00  0.00  0.00
old  woman  remains  in  serious  condition  ,  according  to  @entity7  .  the  driver  has  been
0.00  0.00   0.00     0.00  0.00     0.00       0.00  0.00     0.00  0.00      0.00  0.00  0.00    0.00  0.00
arrested  and  is  under  investigation  by  the  @entity24  .  "  from  the  investigation  and  first
0.00      0.00  0.00  0.00  0.00           0.00  0.00  0.02       0.00  0.00  0.00  0.00           0.00  0.00
findings  ,  there  is  a  strong  suspicion  that  we  're  talking  about  a  terror  attack  ,
0.00      0.00  0.00  0.00  0.00  0.00    0.00       0.00  0.00  0.00  0.00     0.00   0.00  0.00    0.00    0.00
"  @entity7  said  .  amid  the  ongoing  investigation  ,  a  magistrate  court  has  issued  a
0.00  0.00   0.00  0.00  0.00  0.00  0.00     0.00           0.00  0.00  0.00       0.00   0.00  0.00     0.00
gag  order  on  details  of  the  incident  .  <EOS>
0.00  0.00   0.00  0.00     0.00  0.00  0.00      0.00  0.00
```

Q path      : **<SOS> the suspect is a 37 - year - old @placeholder from @entity11 , @entity1 police say <EOS>**

Answer: **@entity10**  Predicted: **@entity10**

CNN, Ortho-LSTM, trained with default (tanh) attention (test acc=.536, conicity=.099)

```
P path      : <SOS>      (      @entity2      )      one      @entity1  citizen  was      killed  and      another  injured  in      what
Attentions : 0.00       0.00  0.19          0.00  0.00      0.00      0.01     0.00      0.00    0.02     0.00     0.00    0.02    0.00
police  are      calling  a      suspected  terror  attack  wednesday  night      near      @entity6  .      @entity8
0.00    0.00     0.00     0.00   0.00       0.00    0.00    0.00       0.01       0.00      0.00      0.03   0.00
spokesman  @entity7  said  a  37  -  year  -  old  @entity10  motorist  from  @entity11  struck  two
0.00       0.00      0.03     0.02  0.00  0.02  0.00  0.01  0.00       0.00      0.01  0.00       0.00    0.00
people  standing  at  a  bus  stop  in  the  @entity15  section  of  the  city  .  one  victim  ,
0.00    0.05      0.13  0.06  0.00  0.00  0.15  0.01  0.00       0.04     0.03  0.01  0.00  0.02  0.00  0.00  0.00
identified  by  police  as  @entity18  ,  26  ,  died  at  the  hospital  .  a  20  -  year  -
0.00        0.00  0.00  0.00  0.00       0.00  0.00  0.00  0.00  0.02  0.00       0.00  0.01  0.00  0.02  0.00  0.01
old  woman  remains  in  serious  condition  ,  according  to  @entity7  .  the  driver  has  been
0.00  0.00   0.00     0.00  0.00     0.00       0.00  0.00     0.00  0.00      0.01  0.00  0.00    0.00  0.00
arrested  and  is  under  investigation  by  the  @entity24  .  "  from  the  investigation  and  first
0.00      0.00  0.00  0.00  0.00           0.00  0.00  0.00       0.00  0.00  0.00  0.00           0.00  0.00
findings  ,  there  is  a  strong  suspicion  that  we  're  talking  about  a  terror  attack  ,
0.00      0.01  0.00  0.00  0.00  0.00    0.00       0.00  0.00  0.00  0.00     0.00   0.00  0.00    0.00    0.00
"  @entity7  said  .  amid  the  ongoing  investigation  ,  a  magistrate  court  has  issued  a
0.00  0.00   0.00  0.00  0.01  0.00  0.00     0.00           0.00  0.00  0.00       0.00   0.00  0.00     0.00
gag  order  on  details  of  the  incident  .  <EOS>
0.00  0.00   0.00  0.00     0.00  0.00  0.00      0.00  0.00
```

Q path      : **<SOS> the suspect is a 37 - year - old @placeholder from @entity11 , @entity1 police say <EOS>**

Answer: **@entity10**  Predicted: **@entity10**

Figure 6: Examples of dual input sequence tasks

