# OpenReview forum: "Reproducibility study - Does enforcing diversity in hidden states of LSTM-Attention models improve transparency?"
_ML_Reproducibility_Challenge/2020 — RC2020_

### Official Review · AnonReviewer1 · 2021-03-02

**Rating:** 7
**Confidence:** 3

**Review:**

Reproducibility Summary:
- The summary did a nice job outlining the report.

Scope of reproducibility:
- Makes clear what the scope of the report is, and provides nice justification for why certain parts were left out (i.e. missing data sets).

Code:
- Uses the original code
- Does an extensive look and dissection of the code base to find some issues worth noting. For example:
  - orthogonalization applied to both the P-path and Q-path), which wasn't obvious from the original text. They do a small experiment to test the model's sensitivity to this change.
  - The final prediction is calculated differently than reported
  - Unreported fine-tuning of the embeddings for the two paths independently
  - An odd choice in how the dev set is chosen from the training set.

Communication with original authors:
- Mentioned sending email to clarify parts of the code with no response.

Hyperparameter Search:
- None done in the reproduction, and none done here.

Ablation Study:
N/A

Discussion on results:
- Did a nice job raising concerns about the original paper's findings.
- Raised many concerns, gave evidence for why these concerns would be problematic (even testing what the changes would be), and gave concrete steps for improvement.

Results beyond the paper:
- Expand the set of evaluation metrics
- Test claim's generality on different architectures. Specifically, the authors applied the metrics to a Bi-directional LSTM model.

Overall organization and clarity

Overall, the report is well written and relatively clear. Some things I would suggest/some questions:
- How many runs did you do for each experiment? Is this similar to what the original authors did?
- table 6, how were the confidence intervals calculated?
- The sub-section "other attention mechanisms" in section 7 is a bit out of place. Maybe you can remove the this entirely, and add it to the final discussion? Because you don't run more experiments here, it just doesn't seem to fit. If you do run experiments here, this needs to be made much clearer.
- The original paper does not seem to do a parameter sweep of any kind, so this would have been a nice inclusion in the report. Event if it wasn't for all the data sets.
- The original paper chooses the best model (out of a set of unknown size) from the validation accuracy and reports the test accuracy. I'm not apart of the community this paper is aimed at (i.e. NLP/Explainability), so I'm not sure how common this is as a practice. Usually, I would prefer to see multiple runs and either a median or average reported with confidence intervals. Of course this can be ignored if this isn't standard practice in this community.

Again. I think this report is well put together and the biggest weakness is the lack of hyperparameter search and ambiguity around how the models were reported (i.e. best of, mean, median) w/o confidence intervals.


**Familiar With The Original Paper:**

I have read the original paper

**Reproducibility Summary:**

Report has summary

---

### Official Review · AnonReviewer4 · 2021-03-08
**Stellar report! Goes above and beyond, and very well written!**

**Rating:** 9
**Confidence:** 5

**Review:**

* Reproducibility Summary

  The report contains a well-defined and articulate reproducibility summary as prescribed by the challenge.
* Scope of reproducibility

  The report contains well-defined scope involving two central claims of the original paper - attention weights not being faithful in plausible explanations in LSTM, and methods to reduce the conicity in order to increase the plausibility of the explanations.
* Code: whether reproduced from scratch or re-used author repository.

  Authors provide their own codebase link, which consists of the code re-used from the original repository. The codebase is well structured with proper README in the appropriate places.
* Communication with original authors

  The report mentions that they have contacted the original authors, but they did not hear back from them. This is unfortunate but sadly happens quite frequently. I applaud the author's effort to reach out to the original authors despite the no reply.
* Hyperparameter Search

  It does not appear that the authors performed an additional hyperparam search than what was reported in the original paper.
* Ablation Study

  The authors compute several extra experiments as part of the ablation of the original work. They add a new evaluation method to clarify the conclusions of the original paper further using LIME. This is a splendid idea, and the correlation results with Pearson correlation (funny it's a correlation of a correlation!) and JS divergence shows the need for such study. The results are mixed, as the main selling point of the paper (Orthogonality and Diversity) does not correlate well with LIME. I would be curious to hear from the authors if they read this review.
  The authors also tested for generalization using Bidirectional LSTM to test the author's claims further (and add a note on why other mechanisms, such as Transformers, are not straightforward to evaluate in the same setting). The authors find the proposed methods do not unconditionally improve the explanations. These kinds of cross-architecture robustness experiments add tons of value to the original paper, and I commend the authors for doing the same.
* Discussion on results

  The report provides a clear and concise discussion of their findings. The authors provide faithful results, both of which experiments worked and which did not. The authors summarized their findings on the original paper and conclude Orthogonal LSTM does clearly leads to lower conicity than Vanilla LSTM, however, the results are mixed. Table 4 is a great summary, clearly defining how each of the claims is supported or not by their reproducibility study.
* Recommendations for reproducibility

  The report goes above and beyond to conduct a thorough code review of the original paper, which is a stellar contribution to both reproducible research and to the understanding of the code provided by the original paper. The authors further provide ample discussion and conclude the benefits of orthogonality and diversity training are more relevant for simpler tasks.
* Overall organization and clarity

  The report does not have any significant typos. It is well organized into appropriate sections.


**Familiar With The Original Paper:**

I have read the original paper

**Reproducibility Summary:**

Report has summary

---

### Decision · Program_Chairs · 2021-03-31

**Decision:**

Accept

**Comment:**

Selected for ReScience-C Journal Publication.

This paper thoroughly examines the the work it aims to build upon, and finds some of the core claims from the original paper were not reproducible. The authors of this reproduction examine the public codebase and describe some potential sources of irreproducibility; we expect this reproduction will be useful to those interested in building upon this work in the future.